# Twin Threats in Digital Workplace: Technostress and Work Intensification in a Dual-Path Moderated Mediation Model of Employee Health

**DOI:** 10.3390/ijerph22121856

**Published:** 2025-12-12

**Authors:** Muhammad Jawwad Nasir Malik, Mubashar Ali, Asad Malik, Shamir Malik

**Affiliations:** 1Faculty of Business and Management Sciences, The Superior University, Lahore 54600, Pakistan; 2Fast School of Management, National University of Computer and Emerging Sciences (FAST NUCES), Islamabad 44000, Pakistan

**Keywords:** technostress, work intensification, IT strain, emotional exhaustion, health harm, user satisfaction, organizational support and digital well-being

## Abstract

**Highlights:**

**Public Health Relevance: How does this work relate to a public health issue?**

**Public Health Significance: Why is this work significant to public health?**

**Public Health Implications: Key implications for practitioners, policymakers, and researchers.**

**Abstract:**

This study investigates how technostress and work intensification jointly influence employee health harm through two distinct stressor-strain pathways within Pakistan’s manufacturing sector. The proposed model specifies two mechanisms, (1) technostress induces IT strain that contributes to health harm, moderated by user satisfaction; and (2) work intensification heightens emotional exhaustion that similarly leads to health harm, moderated by perceived organizational support. Grounded in Conservation of Resources (COR) theory, the framework explains how cumulative digital and organizational demands deplete employee resources, amplifying both psychological and physical harm. A cross-sectional quantitative design was employed, utilizing a structured self-administered questionnaire administered to mid and senior-level employees across manufacturing firms. A total of 252 valid responses were analyzed through Partial Least Squares Structural Equation Modeling (PLS-SEM) using Smart PLS 4. Results revealed that both IT strain and emotional exhaustion significantly mediated the effects of technostress and work intensification, respectively, on health harm. Moreover, user satisfaction significantly moderated the IT strain-health harm relationship, indicating that higher satisfaction with digital tools mitigates the adverse impact of technological stress. Similarly, organizational support weakened the association between emotional exhaustion and health harm, underscoring its protective role in high-pressure work settings. This study offers theoretical advancement by integrating fragmented stressor-strain models and offers practical recommendations to foster digital well-being and supportive organizational work cultures in evolving industrial contexts.

## 1. Introduction

Digital transformation is intensifying psychological pressures in workplaces, raising concerns about employee well-being [1]. Although, globally, digitalization offers substantial efficiency gains, it simultaneously heightens technostress, emotional exhaustion, and health harm as employees struggle with constant connectivity, system complexity, and data or information overload [2]. Contemporary organizations increasingly adopt automation, robotics, cloud platforms, and Artificial Intelligence (AI) to enhance performance and agility. Manufacturing, long a cornerstone of global economic progress, has advanced most rapidly toward Industry 4.0, with Germany, China and South Korea pioneering smart factories to promote innovation and sustainability [3]. While these technological advancements improve efficiency, they also introduce continuous connectivity, algorithmic monitoring, and accelerated work cycles that may exhaust employees [4]. Psychological strain often emerges when digital systems evolve and implemented faster than employees can adapt, generating tension between productivity and workforce well-being [5,6].

Pakistan’s manufacturing sector, contributes approximately 12.4% to GDP and serves as a foundation for exports and employment. The sector faces growing pressure to digitize under the Digital Pakistan Vision, while coping with limited infrastructure, energy shortages, and inconsistent industrial policy. Many factories rely on outdated machinery, low IT penetration and insufficient technical training. As a result, employees are forced to navigate through unfamiliar systems and manage multitasking demands without adequate support. Cognitive and emotional strain is common, exacerbated by top-down technology rollouts, weak change management, and toxic workplace dynamics. These challenges are especially pronounced in SMEs, which often lack structured mechanisms for managing digital transitions.

Health harm manifesting in anxiety, sleep disruption, burnout and hypertension, has become a major concern in the digital work era. As technology adoption accelerates without adequate safeguards, adverse health outcomes are increasingly reported. Two key predictors, technostress and work intensification, act as primary drivers of this problem. Technostress arises from complex, invasive, or rapidly evolving information systems that overwhelm cognitive resources [7]. Work intensification refers to increasing performance demands placed on employees without proportional resource support [8]. These stressors lead respectively to IT strain, defined as persistent digital overload, and emotional exhaustion, characterized by psychological depletion under chronic demands [9]. Both pathways contribute to severe health consequences when left unaddressed.

Within this context, two mitigating factors, organizational support and user satisfaction, play a critical role in reducing these negative effects. Organizational support provides emotional and structural resources that buffer the impact of stressors, while user satisfaction with digital tools enhances positive engagement and reduces frustration [10]. However, many Pakistani manufacturing firms introduce enterprise technologies without adequate consultation, onboarding, or workload redesign. This often produces frustration, affected performance, and higher absenteeism. Employees are commonly required to navigate complex systems independently while working under strict deadlines, bullying-prone environments, and 24/7 digital accessibility, blurring personal and professional boundaries [5]. Such practices erode morale, intensify stress, and undermine the productivity gains expected from digital technologies.

Despite the practical relevance, several important theoretical gaps persist. Existing studies commonly examine technostress and work intensification separately, with limited attention to their combined effects or testing moderated mediation frameworks in resource-constrained manufacturing environments. Mechanisms linking technostress to IT strain and work intensification to emotional exhaustion remain under-explored, and empirical evidence regarding moderators in low-support environments is scarce [11]. Research has yet to capture concurrent dual resource loss spirals, consistent with COR theory in digitally transforming factories, particularly within Pakistan’s resource limited industrial environment [12,13].

Addressing these gaps will enhance theoretical understanding by integrating organizational and technological stressors into a comprehensive unified stressor-strain outcome framework relevant to digital workplaces [6]. Practically, this study aims to guide managers in designing realistic workloads, implementing user-friendly technologies, and promoting inclusive support systems that safeguard well-being while maintaining productivity. The findings are expected to inform policy-makers in embedding psychological safety, constructive feedback mechanisms, and digital capacity-building initiatives into industrial digitization strategies.

Accordingly, this study proposes a dual-path framework in which technostress increases IT strain and work intensification intensifies emotional exhaustion, both leading to health harm. User satisfaction and organizational support are expected to weaken these effects within Pakistan’s manufacturing sector. Based on these gaps, this study addresses the following research questions:How does technostress influence IT strain among manufacturing employees?How does IT strain translate into health harm?Does user satisfaction moderate the IT strain-health harm relationship?How does work intensification generate emotional exhaustion?Does emotional exhaustion predict health harm?Does organizational support moderate the relationship between exhaustion and health harm?

## 2. Theoretical Foundation

### 2.1. Underpinning Theory/Theoretical Foundation

This study is grounded in the Conservation of Resources (COR) theory, which offers a robust lens for explaining how intensified traditional and technological work demands deplete employee resources and impair health [12]. COR theory suggests that individuals are inherently motivated to acquire, protect and preserve valued resources such as psychological energy, time and social support to maintain well-being and cope with environmental stressors [12,14]. According to this framework, stress occurs when individuals face a threat of resource loss, experience an actual loss, or find that their resource investments fail to generate expected returns, ultimately triggering a loss spiral [3,12]. Within this framework, work intensification represents a traditional stressor that increases task complexity and compresses timelines, thereby heightening performance expectations without corresponding support and leading to emotional exhaustion [12,15]. Simultaneously, technostress including techno-overload, techno invasion and techno complexity, functions as a digital-age stressor that exposes workers to constant digital connectivity and system complexity that drain mental resources [1]. These dual stressors act cumulatively and align with COR theory’s premise that chronic, unmanaged demands erode resource pools over time, especially when resources are not replenished [12,14].

The mediating constructs in this model, emotional exhaustion and IT strain, reflect direct manifestations of the resource depletion process. Emotional exhaustion captures the psychological fatigue resulting from sustained pressure, whereas IT strain reflects overload from unstable or difficult technologies [1]. Consistent with COR theory, these depleted states progress into serious health related consequences, collectively referred to as health harm, which includes anxiety, fatigue and physiological disruption [9,16]. The model also incorporates protective variables, organizational support and user satisfaction as moderators that can mitigate resource loss. Organizational support provides emotional and practical resources that buffer the effect of exhaustion on health outcomes [11,12], while user satisfaction enhances autonomy and perceived control, thereby reducing the effect of IT strain [16]. COR theory’s relevance is further demonstrated in digital settings through constructs such as Technological Work Burnout (TWB), which explains how sustained technostress independently drives exhaustion and health deterioration [9,16]. In sum, COR theory provides a comprehensive framework for understanding how technostress and work intensification initiate resource loss spirals that manifest as IT strain and emotional exhaustion. These stressor-strain mechanisms ultimately lead to health harm unless buffered by resource-protective factors such as user satisfaction and organizational support [12,14,16].

Building upon the Conservation of Resources (COR) theory, this study advances theoretical understanding by modeling work intensification and technostress as concurrent stressor domains. These stressors operate together and drain employees’ emotional and cognitive resources. Most prior applications of COR theory examine resource-loss processes in isolation. In contrast, the present dual-path model demonstrates how traditional organizational demands and digital pressures operate simultaneously. These combined demands create two parallel depletion mechanisms, emotional exhaustion and IT strain. This integration shifts COR theory from a single-resource view framework to a multi domain perspective. It explains how intertwined technological and organizational stressors collectively produce cumulative health harm in digitally transforming manufacturing settings.

### 2.2. Operationalization of Variables

Drawing on the principles of COR framework, this study examines seven constructs. Each construct is treated as a distinct capability or performance outcome. Concise definitions are adapted from established scholarship and aligned with the context of Pakistan’s manufacturing context.

#### 2.2.1. Employee Health Harm

Employee health harm in this study reflects the psychological, physical, social, and economic difficulties or multidimensional adverse effects that arise when workers experience continuous depletion of personal resources under intense work and technological demands. Within the resource-conservation perspective [12], it represents the end stage of a cumulative strain process in which chronic persistent organizational and digital demands limit employees’ ability to recover and maintain resilience. The concept also incorporates insights from negative externality of HRM practices framework, which argues that performance-driven management strategies can unintentionally generate harmful health outcomes.

Health harm is operationalized through three validated dimensions. The first dimension, work restriction for health improvements, captures how rigid work schedules and excessive job demands prevent employees from engaging in health-promoting activities, such as physical activity or social interaction [17]. The second dimension, risk factors for negative health, includes early indicators such as emotional strain and chronic fatigue, which signals heightened vulnerability to long-term deterioration. The third dimension, side effect harm, denotes unintended behavioral and physiological outcomes such as disrupted sleep, increased consumption of stimulants, and elevated cardiovascular strain.

These dimensions align with the biopsychosocial model [18] and occupational health psychology perspectives [19], which integrate psychological outcomes (e.g., anxiety, burnout), physical conditions (e.g., fatigue, illness), social challenges (e.g., strained relationships), and economic implications (e.g., reduced productivity, medical expenses) into a holistic view of employee health outcomes. Evidence from workplace and sports literature shows that toxic leadership, high-pressure environments, and unrealistic performance expectations diminish both well-being and satisfaction.

In digital-intensive environments, TWB and IT strain represent key expressions of health harm. These outcomes involve mental fatigue, emotional withdrawal, and reduced capability to function effectively [13]. These conditions emerge when technostress and work intensification persist without adequate buffering, ultimately leading to prolonged psychological and physiological impairment.

Accordingly, employee health harm is defined in this study in line with Mariappanadar (2016) [6] as a multidimensional construct encompassing the psychological, physical, social, and behavioral consequences of prolonged exposure to work intensification and technostress, particularly when these stressors result in sustained personal resource depletion.

#### 2.2.2. Work Intensification

Work intensification is defined as a condition where employees face escalating work demands, increased workloads, and compressed timelines requiring them to complete more tasks at a faster pace, often extending their working hours [20]. It also includes the expectation to swiftly adapt to evolving roles and build new competencies to meet dynamic performance standards.

This phenomenon is compounded by toxic workplace dynamics such as narcissistic leadership, bullying and exclusion, which drain emotional resources and intensify psychological strain [21]. Additionally, perceived organizational politics worsen job dissatisfaction and emotional burden [22].

In a digital context, persistent connectivity and information overload amplify time pressure and disrupt work-life balance, extending the effects of traditional intensification into modern work settings [16,23,24,25]. These compounded demands erode employees’ resources [12], leading to emotional exhaustion and subsequent health harm. Recent Industry 4.0 evidence shows that automation, cyber-physical systems, and digitally integrated workflows have intensified both role overload and time-pressure demands in manufacturing sectors, highlighting the importance of studying work intensification during digital transformation contexts [26].

For this study, the definition of work intensification is adopted from Boxall and Macky (2014), who conceptualize it as the intensification of work pace and effort, driven by heightened organizational expectations and limited resources [20].

#### 2.2.3. Employee Exhaustion

Employee exhaustion refers to the emotional cognitive and physical depletion experienced by individuals exposed to persistent work demands, toxic organizational dynamics, and technological stressors. It represents a multidimensional strain characterized by demotivation, chronic fatigue, impaired focus, and disengagement from work, especially when recovery opportunities are limited.

Exhaustion has been conceptualized as the depletion of energy resulting from prolonged emotional labor and psychological strain. It is often measured using instruments such as the Maslach Burnout Inventory (MBI) and the Oldenburg Burnout Inventory (OLBI). The condition arises not only by excessive workloads and relentless expectations but also from toxic environments marked by harassment, bullying and social exclusion [20,21,27,28]. These factors erode mental and physical resources and accelerate burnout [12,29].

In digital workplaces, exhaustion is intensified by technostress factors such as techno invasion, techno insecurity and techno-overload [16,30,31]. Constant digital stimuli create information overload, while multitasking across platforms limits recovery and heightens cognitive fatigue [24,25]. These patterns replicate and amplify traditional forms of work intensification.

These outcomes reflect broader resource-depletion principles [12] which explain that exhaustion occurs when demands consistently exceed the individual’s available personal resources. In this study, employee exhaustion is defined in line with Gong et al. (2021) as the cumulative depletion of emotional, cognitive and physical energy triggered by persistent workplace stressors [32].

#### 2.2.4. Organizational Support

Organizational support refers to employees’ perceptions that their organization values their contributions and genuinely cares about their well-being [11]. It encompasses both structural mechanisms such as workload management, training and feedback systems and emotional resources that help employees navigate workplace stress [33]. These forms of support are especially critical in environments characterized by increased work intensity and rapid technological change [34].

The role of organizational support becomes particularly salient when employees experience emotional exhaustion due to continuous pressure. Researchers emphasize that emotionally fatigued individuals depend on leadership engagement to preserve their functionality and effective managerial involvement yields significant psychological benefits under strain [22]. Similarly, several studies demonstrate that digital workload pressures and intensified work demands significantly affect employee well-being and performance [1,11,20].

Informal support, including peer encouragement and interpersonal reassurance, also plays a vital role in mitigating distress, especially within toxic work cultures marked by exclusion, bullying, or harassment [35]. These social resources often prove more impactful than formalized policies in addressing emotional needs and sustaining resilience [11].

In this study, the definition by Wang et al. (2020) is adopted to conceptualize organizational support as a vital resource-enhancing condition that helps buffer the effects of emotional exhaustion [11]. In line with a resource-preservation perspective [12], it is positioned as a key moderating variable that replenishes depleted emotional reserves and disrupts the progression towards severe health harm [16].

#### 2.2.5. Technostress

Technostress refers to the psychological strain experienced by employees when workplace technologies become excessive, invasive, or overly complex, thereby exceeding their coping capacities [7]. The construct centers on a set of negative digital stressors, commonly termed techno distress creators, that intensify cognitive load and emotional fatigue by disrupting task flow, work boundaries, and psychological resilience [16]. These stressors are grounded in a resource-depletion perspective [12], which suggests that the persistent loss of emotional and cognitive resources leads to strain and eventual health decline.

Five techno distress creators are widely recognized in Information Systems (IS) literature. Techno-overload occurs when digital tools increase task volume and accelerate work pace beyond manageable limits [7]. Techno invasion disrupts work-life balance by making employees perpetually available, often intruding into personal time [36]. Techno complexity creates emotional strain by requiring continuous learning and adjustment to new systems, which undermines confidence and heightens frustration [37,38]. Techno insecurity arises from anxiety about job displacement due to digital skill gaps or automation [39]. Lastly, techno uncertainty is driven by constant changes and updates in digital systems, keeping employees in a state of adaptation fatigue [40].

While some digital stressors may have enabling effects, termed techno eustress [41], this study focuses exclusively on the cumulative negative impact of techno distress creators that erode emotional and cognitive resources. Technostress is thus defined as a multidimensional digital stressor that elevates IT strain and initiates a sustained resource loss trajectory, ultimately impairing employee health and well-being. Recent research further underscores how technostress undermines digital well-being and human sustainability in the context of Industry 4.0 transformations [42,43].

Technostress in this study is defined in line with Tarafdar et al. (2024) [44], who conceptualize technostress as a function of interrelated distress creators that push users beyond psychological limits. Accordingly, technostress in this study is modelled as a co-digital stressor that underlies IT strain and contributes indirectly to health harm through sustained emotional and mental depletion.

#### 2.2.6. IT Strain

IT strain in the study is conceptualized as the immediate psychological and physical exhaustion experienced by employees due to prolonged exposure to techno stressors embedded within digital work environments [13]. It captures emotional fatigue, cognitive overload, and burnout that arise when Information Systems (IS) demands consistently exceed an employee’s coping capacity [16].

This construct represents the most direct and immediate manifestation of digital stress and acts as a psychological mechanism that links technology-related stressors to deteriorating well-being [16]. The exhaustion associated with IT strain results from continuous exposure to negative stimuli, such as frequent alerts, multitasking pressures, and blurred work-life boundaries [13]. These stressors reduce the opportunity for recovery, ultimately leading to psychological disengagement and fatigue.

The conceptual foundation of IT strain is anchored in a resource-erosion framework, which speculates that chronic loss of emotional and cognitive resources, without sufficient replenishment, leads to psychological strain and eventual health harm [12]. Prolonged use of technology under high-pressure conditions contributes to emotional exhaustion, detachment and avoidance behavior [45].

Moreover, empirical findings demonstrate that IT strain significantly undermines the user’s ability to recover, disturbs work-life balance, and contributes to burnout and TWB [23]. In this study, IT strain is defined as the cumulative emotional and cognitive depletion caused by sustained digital overload, complexity, and uncertainty in technology-intensive work settings.

#### 2.2.7. User Satisfaction

User satisfaction in this study refers to the employee’s affective and cognitive evaluation of the Information Systems (IS) they interact with, encompassing perceptions of usefulness, reliability, ease of use, and alignment with their job tasks [39]. It is widely regarded in IS literature as a surrogate indicator of system success, reflecting the degree to which digital tools fulfill user needs and enhance task performance [46].

In digital workplaces, user satisfaction serves as an essential resource that can help employees manage the pressures arising from prolonged and intensive technology use. When users perceive their systems as supportive, responsive, and aligned with their workflow, they are more likely to maintain emotional stability and psychological resilience, which in turn reduces the harmful effects of IS-induced strain [39].

The dual-factor framework of technostress further explains that, while certain digital conditions generate distress (techno distress creators), others, such as techno enrichment and techno autonomy, can enhance motivation and well-being by increasing user satisfaction [39]. In supportive environments where employees feel digitally competent and autonomous, satisfaction levels rise, creating a positive feedback loop that protects against emotional exhaustion.

This interpretation aligns with a resource-protection viewpoint, which views user satisfaction as a psychological asset that enables employees to conserve emotional and cognitive resources, even under persistent digital demands [12]. A high degree of satisfaction with information systems reduces perceived pressure, supports positive coping, and shields against strain escalation.

Accordingly, for this study, user satisfaction is defined as “a positive emotional and cognitive assessment of workplace information systems that reflects users’ belief that digital tools are effective, supportive and aligned with work tasks” [46]. This adopted definition captures the construct’s dual nature as both an outcome of system quality and a psychological resource that influences user experience under digital stress conditions.

### 2.3. Relationship Between Variables

#### 2.3.1. Relationship Between Work Intensification and Employee Exhaustion

Work intensification, defined by increased workload, compressed timelines, and elevated performance expectations, places employees under sustained pressure to work faster, multitask and continuously upskill within limited timeframes. Such persistent demands frequently exceed an individual’s coping capacity, resulting in both psychological and physical exhaustion [47].

This strain is further exacerbated by toxic workplace conditions, including bullying, harassment, ostracism and hostile supervision, which heighten emotional pressure and rapidly deplete psychological resources [21]. Similarly, perceived organizational politics intensify internal tension and instability, contributing to the onset of exhaustion [22].

The COR theory [12] and the Job Demands-Resources (JD-R) model [48], both support this relationship by asserting that ongoing job demands progressively erode employees’ emotional and cognitive reserves, leading to disengagement and burnout [9,12,16]. The JD-R framework is referenced here as a complementary lens to COR theory, emphasizing how job demands and available resources interact to explain strain outcomes, thereby reinforcing the dual-path logic adopted in this study.

**Hypothesis** **1** **(H1).**
*Work intensification has a significant positive effect on employee exhaustion.*


#### 2.3.2. Relationship Between Employee Exhaustion and Employee Health Harm

Employee exhaustion, characterized by emotional, cognitive, and physical depletion, is strongly associated with adverse health outcomes. Sustained exhaustion impairs emotional regulation, reduces psychological stability, and contributes to diminished well-being, cognitive dysfunction, and behavioral symptoms such as social withdrawal and distress [48]. The llliterature reveals that prolonged emotional depletion, particularly under efficiency-driven HRM regimes, leads to multidimensional harm across psychological, social, and physical domains.

Additional studies confirm that chronic stress exposure heightens anxiety, induces somatic illnesses, and weakens psychological resilience [49]. In digital work environments, the constant psychological demands of adapting to AI systems further amplify exhaustion and reduce mental health and work engagement, consistent with the observation that AI-driven workplaces can heighten emotional exhaustion and anxiety among employees [50]. This often culminates in TWB, which impairs well-being, decreases productivity, and accelerates disengagement [51].

This relationship aligns with the resource-drain perspective [12], which posits that ongoing resource loss without replenishment results in strain and eventual harm, and with the negative externality perspective, which highlights the hidden health costs of demanding digital work environments.

**Hypothesis** **2** **(H2).**
*Employee exhaustion has a significant positive effect on employee health harm.*


#### 2.3.3. Relationship Between Technostress and IT Strain

Technostress arises from techno distress creators such as techno-overload, techno invasion, techno complexity, and techno insecurity, which place persistent cognitive and emotional demands on employees [1,7]. These conditions compel individuals to process excessive information, navigate complex systems, and remain digitally connected beyond regular work hours, disrupting personal boundaries and heightening fears of job insecurity [1]. This sustained exposure depletes emotional and cognitive resources, resulting in psychological strain and burnout-like symptoms [13]. Repeated exposure to such demands without resource replenishment triggers the accumulation of strain [12]. Empirical studies confirm that techno-overload, techno invasion, and techno complexity intensify multitasking burdens and cognitive fatigue, leading to elevated IT strain, and further demonstrate that techno invasion and overload are significantly linked to digital fatigue and emotional exhaustion.

**Hypothesis** **3** **(H3).**
*Technostress has a significant positive effect on IT strain.*


#### 2.3.4. Relationship Between IT Strain and Employee Health Harm

IT strain refers to the psychological and emotional depletion resulting from prolonged and excessive use of information systems, particularly when digital demands consistently exceed employees’ adaptive capacity [13]. These demands contribute to fatigue, anxiety, and emotional burnout, impairing psychological functioning and signaling, deteriorating well-being. In digitalized work environments, this strain is further intensified by AI-induced workloads that heighten both psychological and physical stress responses, thus exacerbating employee health vulnerabilities. Sustained IT strain also leads to TWB, a syndrome characterized by cognitive overload, emotional exhaustion, and disengagement from digital work tasks, which collectively undermine mental health and overall productivity [23]. Taken together, these insights underscore that IT strain is a critical early indicator of harm, functioning as the key conduit through which technostress evolves into broader psychological and physiological damage [13].

**Hypothesis** **4** **(H4).**
*IT strain has a significant positive effect on employee health harm.*


#### 2.3.5. Relationship Between Work Intensification, Employee Exhaustion, and Health Harm

Work intensification, marked by rising job demands, compressed timelines, and constant performance pressure, acts as a powerful organizational stressor that drains employees’ psychological and physical resources. This depletion leads to employee exhaustion, which operates as the key mechanism through which intensified workloads contribute to adverse health outcomes. Organizational politics, often embedded within high-intensity environments, intensify emotional strain and competitiveness, thereby increasing the risk of health harm through resource exhaustion rather than direct effects [22].

Empirical studies support this meditating pathway. Gong et al. (2021) found that elevated work demands raise emotional exhaustion, which then diminishes psychological well-being [32]. Similarly, researchers demonstrated that work intensification impacts performance and well-being indirectly via mediating variables like exhaustion and creativity, highlighting the internal psychological toll as the transmission channel [52].

The COR theory [13] underpins this sequence by asserting that prolonged demands erode core personal resources, such as energy and resilience, which, once depleted, result in exhaustion that subsequently leads to health deterioration [12]. Researchers emphasized that exhaustion resulting from unsustainable work systems causes harm across psychological, social and physiological domains.

This resource-conservation mechanism is also observed in technology-driven environments, where continuous digital demands heighten exhaustion and gradually result in TWB and broader health decline [16]. Thus, whether arising from traditional or digital sources, exhaustion consistently emerges as the mediating link between work intensification and health harm.

**Hypothesis** **5** **(H5).**
*Employee exhaustion mediates the relationship between work intensification and employee health harm.*


#### 2.3.6. Relationship Between Technostress, IT Strain, and Health Harm

The pathway from technostress to health harm operates through a critical psychological mechanism, IT strain, which translates digital demands into deteriorating employee well-being [44]. Technostress, triggered by factors such as techno overload, complexity, and invasion, leads to the depletion of emotional and cognitive resources, which initially manifests as IT strain—a state of psychological fatigue and emotional exhaustion [12,13]. Sustained strain builds into broader physical and psychological harm when employees are unable to replenish lost resources [12].

This mediating pathway is supported by empirical evidence. It has been established that that unmanaged strain from AI-induced job pressures results in health issues that erode the intended benefits of technological integration [53]. Similarly, exposure to techno distress factors leads to escalating IT strain, which culminates in health damage when left unaddressed [16]. Researchers further identify TWB as the end state of this progression, characterized by fatigue, cognitive dysfunction, and psychological disengagement stemming from accumulated digital strain. Thus, IT strain plays a central mediating role in the stress-response chain, illustrating how technostress indirectly leads to employee health harm by exhausting essential psychological resources [12].

**Hypothesis** **6** **(H6).**
*IT strain mediates the relationship between technostress and employee health harm.*


#### 2.3.7. Relationship Between Organizational Support, Employee Exhaustion, and Health Harm

Organizational support plays a critical role in mitigating the health-damaging effects of emotional exhaustion by providing employees with reassurance and emotional care through supportive leadership [22,54]. When employees feel psychologically safe, they are better able to manage strain associated with high-pressure environments [22]. Perceived organizational support, defined as the belief that the organization genuinely values employees’ well-being, amplifies this effect by offering both emotional and instrumental resources [11]. These include practical tools such as training, workload assistance, and emotional encouragement that help restore psychological balance [34].

Such resources allow employees to better cope with emotional exhaustion and limit the translation of this exhaustion into psychological or physical harm [11,35]. Social support mechanisms, including interpersonal reassurance from peers, have also been shown to reduce psychological distress under toxic or isolating conditions [34,35]. These findings align with the stress-buffering function of support systems, where the availability of care and assistance moderates the relationship between job strain and health decline [11]. In this study, organizational support is thus treated as a moderator that weakens the direct impact of employee exhaustion on health harm.

**Hypothesis** **7** **(H7).**
*Organizational support moderates the relationship between employee exhaustion and health harm such that the relationship is weaker when organizational support is high.*


#### 2.3.8. Relationship Between User Satisfaction, IT Strain, and Health Harm

User satisfaction, defined as a favorable perception of an information system’s usefulness, ease of use, and relevance to work tasks, reduces the psychological burden caused by IT strain. When satisfaction is high, employees interpret digital demands more positively, thereby limiting the emotional depletion typically associated with IT strain [39]. Empirical findings confirm that employees with higher system satisfaction report fewer negative effects on well-being despite experiencing digital strain.

Satisfaction, coupled with resilience and optimism, weakens the impact of AI-related strain on mental health. Satisfaction reduces the psychological effects of workplace digital stressors. In digitally intensive contexts, techno enrichment and techno mastery are positively associated with user satisfaction and lower risk of health harm.

This moderating effect aligns with the COR theory, where satisfaction functions as a resource gain that offsets the losses caused by strain [12]. Thus, user satisfaction acts as a psychological buffer that weakens the link between IT strain and health harm.

**Hypothesis** **8** **(H8).**
*User satisfaction moderates the relationship between IT strain and employee health harm such that the relationship is weaker when user satisfaction is high.*


### 2.4. Theoretical Framework

This study employs the COR theory [12,14] to explain how organizational and technological stressors deplete employees’ psychological, emotional, and physical resources, ultimately resulting in health harm. It suggests that individuals experience stress when their resources are threatened, lost, or insufficiently replenished after expending effort [12]. In traditional settings, work intensification is characterized by elevated demands and limited recovery, leading to systematic exhaustion of employee resources and resulting in emotional depletion and burnout [12,20,47]. Simultaneously, the digital dimension of workplace stress is captured through technostress, which arises from techno distress creators such as techno-overload, techno invasion, techno complexity, and techno insecurity. These stressors escalate task complexity, blur work-life boundaries, and foster fears of technological redundancy [16,36,44]. As a result, employees develop IT strain, a psychological state marked by mental fatigue, digital exhaustion, and cognitive overload due to continuous interaction with demanding information systems [13,44]. Both pathways, work intensification leading to employee exhaustion and technostress leading to IT strain, converge toward employee health harm. This harm includes psychological, physical, and social deterioration such as anxiety, depression, and burnout [18,19]. These outcomes are consistent with the negative externality of HRM perspective, which asserts that internal-efficiency-focused HRM systems impose hidden social costs, particularly in the form of health damage, on employees. To mitigate these effects, the framework incorporates two key moderators. At the organizational level, perceived support offers emotional and instrumental resources that buffer the link between exhaustion and health harm [11,35]. At the technological level, user satisfaction, based on ease of use, usefulness, and goal alignment, buffers the link between IT strain and health harm by fostering positive system experiences [44]. This framework also reflects the dual-factor stress theory, which contrasts techno distress with techno eustress conditions such as techno mastery and techno enrichment that enhance satisfaction and reduce the escalation of strain [41]. These insights are further complemented by self-determination theory and proactive behavior literature, which emphasize that autonomy and proactive engagement improve psychological outcomes in high-demand environments [55,56]. The theoretical framework (Figure 1-proposed model) thus comprehensively integrates traditional and digital stressors, their mediating mechanisms, and the protective roles of organizational and technological buffers in shaping employee well-being.

## 3. Research Methodology

This study adopts a positivist research philosophy, emphasizing objective measurement, empirical testing, and statistical analysis to examine causal relationships among defined constructs [57]. It is well-suited for investigating how work intensification and technostress contribute to IT strain and employee exhaustion, ultimately leading to health harm, and for evaluating the moderating roles of organizational support and user satisfaction. The focus on quantifiable variables and hypothesis testing ensures generalizable insights into stressor-strain dynamics in digitalized workplaces.

Following this paradigm, a deductive reasoning approach was employed, grounded in the COR theory [12]. Hypotheses were derived from theory and tested empirically through structured data collection and quantitative analysis [58]. This enhances theoretical alignment and methodological rigor, supporting both conceptual validation and practical relevance, particularly in manufacturing settings undergoing digital transformation.

A cross-sectional survey design was chosen to collect data at a single point in time, offering snapshot insights into employee psychological states and workplace perceptions [59]. This design is widely used in stressor-strain outcome research and provides advantages of cost-efficiency and wide coverage [60]. A structured, self-administrated questionnaire minimized interviewer bias and allowed respondents the flexibility to answer thoughtfully [61].

The target population included mid-level and senior managers and knowledge workers in Pakistan’s manufacturing industry, individuals with regular exposure to digital systems and performance pressures. They were purposively selected for their ability to provide informed responses on key constructs such as technostress, IT strain, work intensification, exhaustion, organizational support, and user satisfaction [62]. The organization served as the unit of analysis, with each participant acting as a knowledgeable informant, making the approach suitable for evaluating broader digital work practices.

Since this study employed a cross-sectional design, the relationships tested represent associations rather than causal effects. While performing analysis, preliminary data screening and descriptive statistics were conducted in SPSS 27, whereas measurement and structural model assessments were performed using Smart PLS 4 through the Partial Least Squares Structural Equation Modeling (PLS-SEM) approach.

### 3.1. Sampling Strategy

The study adopted a non-probability purposive sampling strategy to gather context- specific insights from employees engaged in digital and performance-related processes within Pakistan’s manufacturing sector [63,64]. Senior HR managers assisted in targeting mid and senior-level respondents, having experienced the constructs under study. This approach was appropriate because only employees at these organizational levels possess direct exposure to digital systems, workflow intensification, and managerial expectations, making them the most suitable respondents for the study’s objectives. A mixed-mode approach was followed: Google Forms enabled digital accessibility [65], while printed questionnaires ensured inclusivity for on-site managers in industrial zones, enhancing response rates and coverage [66]. Of 350 questionnaires, 287 were returned and 252 retained after screening, yielding a 64% valid response rate, adequate for SEM-based analysis [67]. The final sample of 252 responses was deemed adequate for PLS-SEM analysis, as it exceeds the minimum recommended threshold under the 10-times rule and Cohen’s (1992) power criterion (power = 0.80, medium effect size, α = 0.05) [67], consistent with the guidelines proposed by Hair et al. (2017) for models of comparable complexity [62], thereby ensuring sufficient statistical power for the hypothesized mediation and moderation effects.

### 3.2. Measurement of Variables

All constructs in this study were measured using validated multi-item scales sourced from peer-reviewed literature to ensure theoretical rigor and measurement reliability [57]. Responses were captured on a five-point Likert scale ranging from 1 (Strongly Disagree) to 5 (Strongly Agree), a format well-suited for capturing attitudinal and perceptual data in organizational research [68]. The dependent variable, Employee Health Harm, was assessed through 14 adapted items [6], with scale verification, yielding a Cronbach’s alpha of 0.943. This construct captures the emotional, psychological, and lifestyle-related health outcomes associated with workplace stress. Sample items include: “I feel ‘down in the dumps’ that nothing cheers me up,” “I feel nervous of late,” and “I frequently have disturbances to normal sleep.” The independent variable Work Intensification was measured using 10 adapted items [20], which were further validated and extended, with a reported Cronbach’s alpha of 0.972. The items reflect two subdimensions: Role Overload and Time Demands. Example items include: “I feel like I have too much work to accomplish in a limited amount of time” and “I often feel rushed due to strict time demands at work.” Employee Exhaustion, modeled as a mediator, was measured using six adapted items and drawn from the exhaustion component of the Maslach Burnout Inventory [69]. A validated scale, yielded a Cronbach’s alpha of 0.969. It captures emotional and physical depletion caused by sustained workload, with items such as: “I feel emotionally drained from my work” and “I feel burned out from my work.” Organizational Support, the moderating variable between exhaustion and health harm, was assessed using four adapted items, reporting a Cronbach’s alpha of 0.784. Items measure perceived care and flexibility, such as: “The organization attaches great importance to my work goals and values” and “The organization provides me enough time to deal with my family matters.” Technostress was measured using 14 items adapted from the Technostress Creators scale [70], with further validation [7]. This construct comprises three dimensions, Techno-Overload, Techno-Invasion, and Techno-Complexity. Sample items include: “I am forced by information technology to work much faster” and “I feel my personal life is being invaded by information technology.” The reported Cronbach’s alpha values in the original study were 0.83, 0.81, and 0.75 for the respective dimensions. IT Strain, which captures the psychological fatigue arising from prolonged IT system usage, was measured using a three-item scale [13], with a Cronbach’s alpha of 0.82. Representative items include: “Working all day, with IT applications and devices is a strain for me” and “I feel burned out from activities that require me to use IT applications and devices.” User Satisfaction, modeled as a moderator between IT strain and health harm, was assessed using a four-item scientific differential scale [71]. Items capture users’ overall satisfaction with IT systems with bipolar adjective pairs, such as “Dissatisfied–Satisfied” and “Frustrated-Contented.” This measure demonstrated strong internal reliability, with a Cronbach’s alpha of 0.89. All items were thoroughly reviewed and pre-tested for clarity and contextual appropriateness within the Pakistani manufacturing setting. Consistent with best practices, all adapted scales demonstrated acceptable internal consistency with Cronbach’s alpha values exceeding the 0.70 threshold [72,73]. This robust measurement framework supports the validity and reliability required for SEM-based hypothesis testing.

Cronbach’s alpha values cited earlier refer to those reported in the original validation studies of the adopted scales. The reliability coefficients computed for this study are presented in Table 2. All α and Composite Reliability (CR) values lie within acceptable thresholds (0.70–0.95), confirming measurement consistency. In line with PLS-SEM recommendations, CR was emphasized as a more appropriate reliability measure for the current model [63].

Both Work Intensification and Technostress were modeled as second-order reflective constructs composed of interrelated subdimensions (Role Overload and Time Demands for Work Intensification; Techno-Overload, Techno-Invasion, and Techno-Complexity for Technostress). The repeated indicator approach was applied to estimate their higher-order latent structure concurrently within the PLS path model.

### 3.3. Scale Reliability and Pre-Validation

All adapted scales used in the study demonstrated acceptable levels of internal consistency, with Cronbach’s alpha values exceeding the recommended threshold of 0.70 in prior literature. The items were also pre-tested to ensure clarity, contextual relevance, and construct validity before full-scale distribution. All analyses were performed on fully completed questionnaires (N = 252); no missing values were imputed or substituted.

### 3.4. Data Normality

Previous research has established that PLS-SEM is a non-parametric approach, hence not affected by abnormal data distribution [63], although the assessment of data normality distribution is considered essential before applying inferential statistics [74]. In this study, data normality was examined through skewness, kurtosis, and histogram plots [75]. The obtained values for all variables were within a defined threshold of −2 to +2, indicating a highly degree of data normality.

### 3.5. Demographics

A total of 287 respondents participated in the survey (35 participants’ data removed as outliers through Mahalanobis distance formula). After data treatment, 252 responses were evaluated. Out of 252 participants, 75.6% (189) were male and 24.4% (63) female with all married. Most respondents were aged from 46–60 years (75.79% (191)), 10% (25) were in the “60 and above” group, 12% (30) were in the “31–45 years” group, and the remaining 2.38% (6) were under 30 years old. Responses related to “length of service” disclosed that 2.8% (7) of employees had 6–10 years’ job experience, 10.71% (27) had 11–15 years, 73.41% (185) had 16–20 years, and the remaining 13.09% (33) had 60+ years in respective organizations. In terms of education, 19.05% (48) had a Bachelor’s degree, 75.79% (191) a Master’s, and 5.16% (13) had a doctorate degree. In terms of employment, 63.5% (160) were managers and 36.5% (92) were mid-level employees. Respondents’ demographic characteristics are given in Table 1 and Figure 2.

### 3.6. Data Analysis and Structural Equation Modeling (SEM)

The theoretical framework’s parametric requirements were assessed, including measurement model assessment, which evaluated the reliability of items and the validity of the scale, followed by structural model assessment for hypothesis testing. The Partial Least Squares Structural Equation Modeling (PLS-SEM) technique was utilized via Smart PLS 4 software, given its robustness for analyzing complex models with direct, indirect, and moderating effects [63]. PLS-SEM was adopted for several reasons: it effectively explains and predicts constructs, offers flexibility in framework construction, and does not impose strict data normality or large sample requirements. The PLS algorithm was employed to assess construct validity through the Confirmatory Factor Analysis (CFA) technique, followed by the bootstrapping technique for hypothesis testing. Additionally, RStudio (version 4.3.1) was used to generate statistical visualizations and support the representation of results.

## 4. Results

### 4.1. Measurement Model Assessment (MMA)

In measurement models (outer models), the relationship between items and variables is established [76]. Convergent validity and discriminant validity form the core in measurement model evaluation [77]. In a reflective measurement model, validity and reliability are assessed in relation to a “latent variable” [63]. The assessment employed the Confirmatory Factor Analysis (CFA) technique to ascertain the reliability of items (internal consistency) and validity of the scale (convergent and discriminant validity).

#### 4.1.1. Convergent Validity (CV)

Convergent Validity (CV) was assessed using factor loadings, Average Variance Extracted (AVE), and Composite Reliability (CR). As recommended in prior methodological literature [78], a minimum loading of 0.70 and an AVE threshold of 0.50 were applied, with all constructs exceeding these thresholds. Composite reliability values also surpassed recommended levels of 0.70, confirming internal consistency [63]. For second-order constructs (Work Intensification and Technostress), the repeated indicator approach was applied, and AVE and CR values were calculated manually using the established technique. As shown in Table 2 and Figure 3, all results met acceptable parameters.
ijerph-22-01856-t002_Table 2Table 2CV (Validity and Reliability of Constructs).ConstructsItemsLoadingAlphaCRAVEEXEX10.7380.880.8830.626
EX20.797



EX30.83



EX40.796



EX50.764



EX60.818


HHHH10.6640.9430.9460.577
HH100.797



HH110.801



HH120.793



HH130.756



HH140.652



HH20.712



HH30.637



HH40.76



HH50.765



HH60.866



HH70.848



HH80.782



HH90.758


ITSITS10.8370.7990.80.713
ITS20.851



ITS30.846


OSOS10.8690.8810.8930.736
OS20.863



OS30.87



OS40.83


RORO10.9040.9420.9420.775
RO20.874



RO30.865



RO40.875



RO50.872



RO60.89


TCTC10.8730.908 0.91 0.732 
TC20.814



TC30.874



TC40.826



TC50.888


TDTD10.8080.8530.8570.695
TD20.85



TD30.866



TD40.809


TITI10.9070.920.9240.806
TI20.892



TI30.873



TI40.918


TOTO10.8270.863 0.8 64 0.647 
TO20.81



TO30.795



TO40.757



TO50.831


USUS10.7820.8670.9250.703
US20.872



US30.828



US40.87


WI

0.9120.8390.725TS

0.8740.7680.526**Note:** AVE denotes Average Variance Extract, CR represents Composite Reliability, and CA stands for Cronbach’s Alpha. EX = Employee Exhaustion; HH = Employee Health Harm; ITS = IT Strain; OS = Organizational Support; RO = Role Overload; TC = Techno complexity; TD = Time Demands; TI = Techno invasion; TO = Techno-Overload; US = User Satisfaction; WI = Work Intensification; TS = Technostress.

#### 4.1.2. Discriminant Validity (DV)

The DV, which signifies the degree of a variable’s distinctiveness from others, was assessed in this study using the heterotrait-monotrait ratio method [79]. The HTMT ratio has been proposed as a preferred method to estimate DV [79]. Earlier techniques such as the Fornell-Larcker criterion and cross-loadings were also considered effective for measuring validity, but they often failed to detect discriminant validity in certain complex cases [80]. To ensure the validity of the test, it is recommended that the HTMT ratio be either below 0.85 (HTMT < 0.85) as suggested by (Clark and Watson 1995, Kline 2011) below 0.90 (HTMT < 0.90) (Gold, Malhotra et al. 2001); if it exceeds a given threshold, then DV is not validated [81,82,83]. In Table 3, all HTMT values are below 0.90, confirming satisfactory discriminant validity. The findings therefore indicated that DV was proven for all variables. To assess DV, HTMT values were computed separately for first and second-order constructs to ensure clarity of results (see Table 3/Figure 4 and Table 4/Figure 5). Figure 6 illustrates the statistical model results.

### 4.2. Structural Model Assessment (SMA)

MMA is a prerequisite for SMA. Following MMA, SMA was pursued to validate the hypothesis by examining path coefficients, t-values, *p*-values, and confidence intervals. Hypotheses were tested using Smart PLS 4 software and the bootstrapping technique.

#### 4.2.1. Path Analysis

In the path analysis, direct and indirect relationships between variables were assessed using the SEM technique, and the results are summarized in Table 5. The analysis confirmed statistical support for all hypothesized relationships. A significant positive association was found between Work Intensification and Employee Exhaustion (β = 0.346, t = 6.089, *p* = 0.00, LL = 0.249, UL = 0.437), thereby supporting H1. Employee Exhaustion was also positively associated with Employee Health Harm (β = 0.608, t = 7.07, *p* = 0.00, LL = 0.482, UL = 0.77), confirming H2. Similarly, a significant and positive relationship was observed between Technostress and IT Strain (β = 0.277, t = 4.146, *p* = 0.00, LL = 0.17, UL = 0.389), supporting H3. IT Strain was also found to be positively associated with Employee Health Harm (β = 0.338, t = 5.724, *p* = 0.00, LL = 0.233, UL = 0.435), thereby confirming H4. Furthermore, Employee Exhaustion was found to mediate the relationship between Work Intensification and Employee Health Harm (β = 0.21, t = 4.735, *p* = 0.00, LL = 0.145, UL = 0.286), providing support for H5. Moreover, IT Strain was shown to mediate the relationship between Technostress and Employee Health Harm (β = 0.094, t = 3.662, *p* = 0.00, LL = 0.057, UL = 0.141), which supports H6. The moderating effect of Organizational Support on the relationship between Employee Exhaustion and Employee Health Harm was also confirmed (β = −0.127, t = 2.393, *p* = 0.009, LL = −0.226, UL = −0.047), thus supporting H7. Lastly, User Satisfaction was found to moderate the relationship between IT Strain and Employee Health Harm (β = −0.108, t = 2.677, *p* = 0.004, LL = −0.172, UL = −0.036), confirming H8. Figure 7 presents the mediation effects and explains that high organizational support weakens the positive effect of exhaustion on Employee Health Harm. Similarly, high User Satisfaction reduces the impact of IT Strain on Employee Health Harm. Additionally, Figure 8 explains the overall SMA model.

#### 4.2.2. Valuation of the Coefficient of Determination (R^2^)

The concept of R^2^, known as the coefficient of determination, is pivotal in Structural Equation Modeling (SEM), representing the proportion of variance in the endogenous construct accounted for by the exogenous constructs. The R^2^ measures the predictive power of endogenous variables, as an alternative means to assess model quality in SEM. Thresholds of 0.19, 0.33, and 0.67 are generally used to classify weak, moderate, and substantial explanatory power, respectively [84]. Table 6 shows R^2^ values for endogenous variables.

#### 4.2.3. Assessment of the Effect Size (f^2^)

The concept of effect size pertains to the influence of exogenous constructs on endogenous constructs through changes in R^2^ values [85]. When a latent variable is removed, the alteration in the R^2^ value quantifies the effect size (f^2^) [86]. Values of 0.02, 0.15, and 0.35 are typically classified as small, medium, and large effects, respectively [87]. Table 7 displays effect sizes for all path coefficients.

#### 4.2.4. Summary of Findings

The structural model results were consolidated to provide an overall evaluation of the hypotheses. As shown in Table 8, all proposed relationships were statistically supported.

## 5. Discussion

All hypothesized relationships (H1–H8) were supported with statistically significant path coefficients and acceptable explanatory power, as reported in Table 6, Table 7 and Table 8. This confirms the robustness of the model and offers a strong empirical basis for the theoretical interpretations.

The findings validated all hypothesized relationships by showing that work intensification significantly drives employee exhaustion, which subsequently heightens health harm (H1, H2). Similarly, technostress was found to increase IT strain, which also contributes to health harm (H3, H4). Mediation analysis further confirmed that exhaustion explains the pathway from work intensification to health harm (H5), while IT strain mediates the relationship between technostress and health harm (H6). Importantly, organizational support emerged as a key moderator that weakens the exhaustion-health harm link (H7), and user satisfaction buffered the effect of IT strain on health harm (H8). Together, these results affirm dual stressor-strain mechanisms consistent with COR theory, while underscoring the protective role of organizational and technological resources in digitally intensive workplaces.

These results align with Tarafdar et al. (2024) [44], who reported that techno overload and techno invasion increase strain and exhaustion in digital workplaces. Similarly, Boxall and Macky (2014) found that intensified workloads predict emotional exhaustion, supporting the present study’s findings within an Industry 4.0 context [20].

Although all hypothesized relationships were statistically significant, the practical relevance varied according to the effect size. Exhaustion demonstrated the strongest predictive effect on health harm (f^2^ = 0.306) followed by IT strain (f^2^ = 0.203) and through a smaller yet meaningful influence of work intensification (f^2^ = 0.136). These variations indicate that exhaustion contributes more substantially to employee health deterioration, underscoring the comparatively stronger influence of psychological strain over technological strain in the dual-stressor model.

### 5.1. Theoretical Implications

This study made significant theoretical contributions by enhancing the understanding of how dual stressors, work intensification and technostress, jointly influence employee health harm within the manufacturing sector of Pakistan. Grounded in COR theory, the findings clarified how organizational demands deplete employees’ emotional resources, resulting in exhaustion, while unmanaged digital demands intensify IT strain. Both pathways independently and interactively led to psychological and physical deterioration [13,88]. By empirically validating the parallel operation of these two stressor-strain mechanisms, the study advanced multi path stressor models, addressing gaps in prior research that often examined these constructs in isolation. The study further contributed by confirming the moderating roles of organizational support and user satisfaction as important buffers against the negative effects of exhaustion and IT strain, respectively [11,16,89]. Additionally, the study unified fragmented literature by integrating technological and organizational stressors into a dual-path conceptualization, particularly within an under-researched context such as Pakistan’s digitally transforming manufacturing sector. Instead of viewing user satisfaction solely as a functional system outcome, this study demonstrated and empirically validated its emotional role as a moderator that shapes how employees internalized technological demands. Likewise, organizational support was positioned as an active protective resource that reduced the progression from exhaustion to health harm. These insights enriched stressor-strain outcome models and established a robust theoretical lens for future investigations into employee well-being in digital work environments.

### 5.2. Practical Implications

From a practical perspective, the study offered actionable guidance for managers, HR professionals, and policymakers engaged in digital transformation. The findings highlighted the need to address work intensification through interventions such as workload calibration, structured recovery time, and supportive supervision to reduce exhaustion related health risks. Simultaneously, the results indicated the importance of integrating user-centered practices into digital implementations, such as ongoing training, intuitive interfaces, and psychological support mechanisms, to mitigate technostress and enhance user satisfaction [13,16]. Organizations may consider introducing digital ergonomics and IT-user training initiatives to minimize technostress and improve digital proficiency. Periodic workload reviews and flexible scheduling could help mitigate work intensification, while psychosocial risk-management measures, such as stress audits and employee support channels, may promote digital well-being. Policymakers might also explore issuing occupational digital well-being guidelines to support healthy technology adoption across industries. Conceptualizing health harm as a measurable organizational risk provides firms with an early-warning indicator for preventing long-term productivity losses, supporting the adoption of sustainable HRM strategies that link technology use with well-being [86,90]. The findings further suggest that shifting from reactive to preventive models of employee care and fostering psychologically safe work environments can help align workforce sustainability with digital competitiveness in the evolving Industry 4.0 landscape.

### 5.3. Limitations

This study, while offering useful insights into how work intensification and technostress affect employee health in Pakistan’s manufacturing sector, has some limitations. The model focused primarily on two stressors, work intensification and technostress, mediated by exhaustion and IT strain and moderated by organizational support and user satisfaction. Other factors, such as self-efficacy, flexible work arrangements, and leadership styles, may also influence health outcomes but were not included [13]. This highlights the need to explore additional predictors for a more comprehensive explanation of the mediating mechanisms.

The research relied on self-reported, cross-sectional data, which limits causal claims. Although steps were taken to minimize method bias, self-reported perceptions may inflate associations among variables, and the design lacks the temporal strength of longitudinal approaches. Furthermore, although validated measures were used, constructs such as health harm may vary across cultures, job roles, or organizational contexts, which limits generalizability [13].

The demographic composition of the sample also poses limitations. As most respondents were male, mid-to-senior-level employees aged 46–60 years, the findings may not fully generalize to younger or non-managerial workers, reflecting the demographic composition of Pakistan’s manufacturing sector.

As purposive sampling targets specific informed respondents, rather than a randomly selected population, the findings may not fully generalize beyond similar managerial groups in Pakistan’s manufacturing sector.

Additionally, cultural and sectoral characteristics, such as hierarchical management practices, limited automation readiness, and evolving national digitalization policies, may further influence how employees perceive and respond to technostress and work intensification. These contextual factors may constrain the extent to which the findings generalize to more digitally mature economies.

### 5.4. Future Research Directions

Future studies should employ longitudinal or time-lagged designs to better capture cause-and-effect relationships and reduce common method bias. Integrating employee responses with objective health indicators, such as attendance records or supervisor feedback or evaluations, would improve measurement accuracy. Researchers may also explore digital traits such as IT mindfulness, digital literacy, and techno eustress, which were not examined in this study but may affect how individuals manage technology related demands and handle technology at work [90]. Since the findings are based on Pakistan’s manufacturing sector, future research should test this model in other industries and cultural contexts. Lastly, individual characteristics such as age, gender, and job experience may also influence how employees respond to stressors, and future studies should investigate how personal and organizational resources interact to shape well-being [22]. Given its comparatively stronger effect on health harm, exhaustion merits particular attention in future research.

## 6. Conclusions

This study examined how technostress and work intensification contribute to employee health harm in Pakistan’s manufacturing sector through the meditating roles of IT strain and emotional exhaustion, moderated by user satisfaction and organizational support. Grounded in COR theory and tested through PLS-SEM, the results confirm that both digital and organizational stressors affect health through distinct psychological mechanisms. Technostress leads to IT strain and health harm, effects that were buffered by user satisfaction. Similarly, work intensification heightened exhaustion and health harm, moderated by organizational support. These findings underscore the importance of managing workloads, improving digital usability, and fostering supportive work environments to safeguard employee well-being and promote sustainable operations in digitally evolving workplaces. Overall, the results highlight that sustaining employee health in the digital era requires balancing technological demands with resource-protective organizational practices that promote long-term well-being and resilience.

## Figures and Tables

**Figure 1 ijerph-22-01856-f001:**
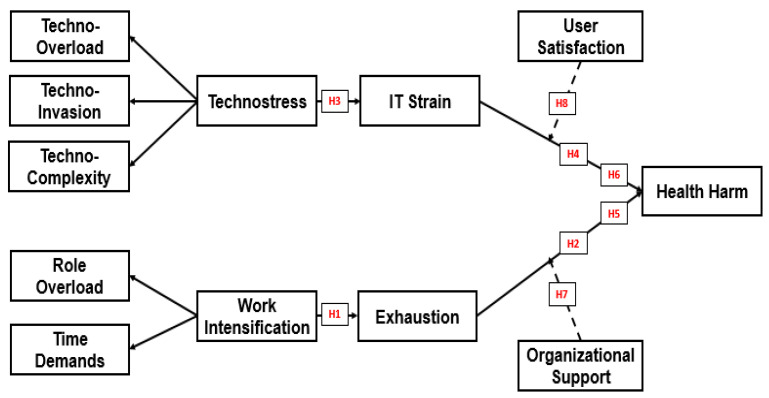
Proposed Model. Solid lines indicate direct and mediating relationships (H1–H4: direct relationship; H5–H6: mediating relationship); dashed lines represent moderating effects (H7–H8).

**Figure 2 ijerph-22-01856-f002:**
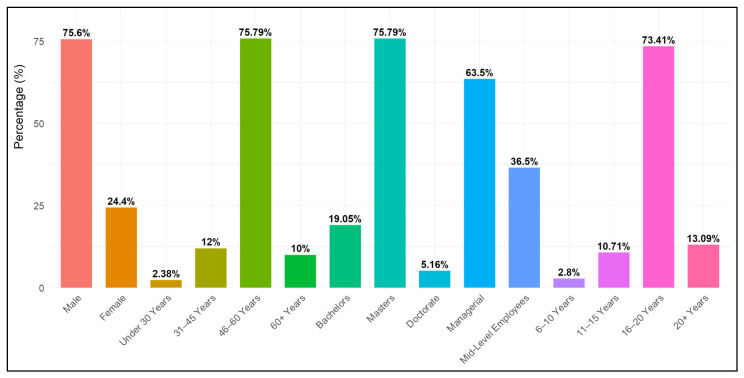
Demographic Distribution of Survey Participants by demographic characteristics (Gender, Age, Education, Employment, and Service length), expressed as percentages of the total sample; N = 252.

**Figure 3 ijerph-22-01856-f003:**
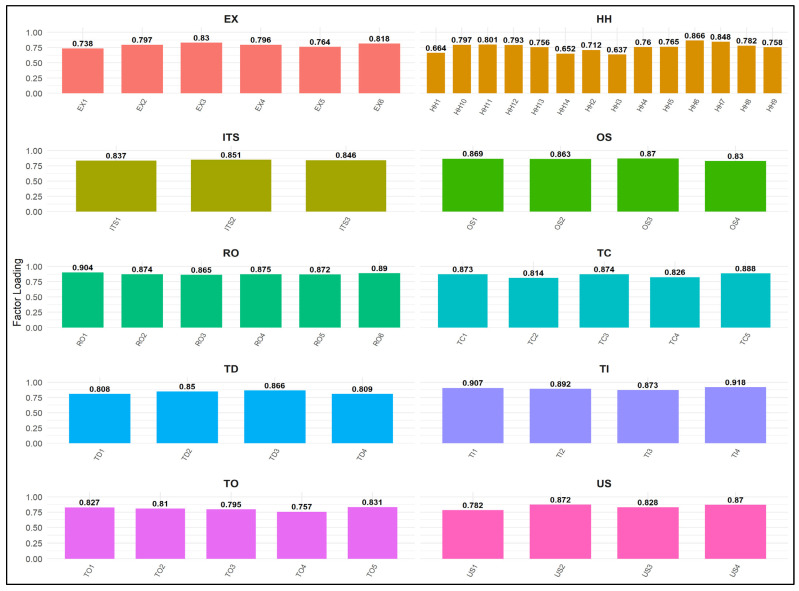
Measurement Model: Item Loadings by Construct: Results showing item loadings across constructs (EX, HH, ITS, OS, RO, TC, TD, TI, TO, US). Reliability statistics for each construct (Cronbach’s Alpha, Composite Reliability, AVE) indicate good internal consistency and convergent validity.

**Figure 4 ijerph-22-01856-f004:**
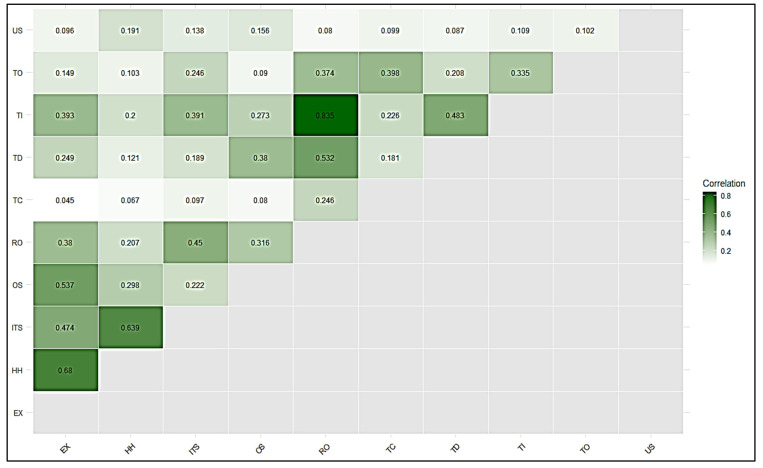
Correlation matrix among first-order constructs. Values represent inter-construct correlations, with stronger associations highlighted in darker shades.

**Figure 5 ijerph-22-01856-f005:**
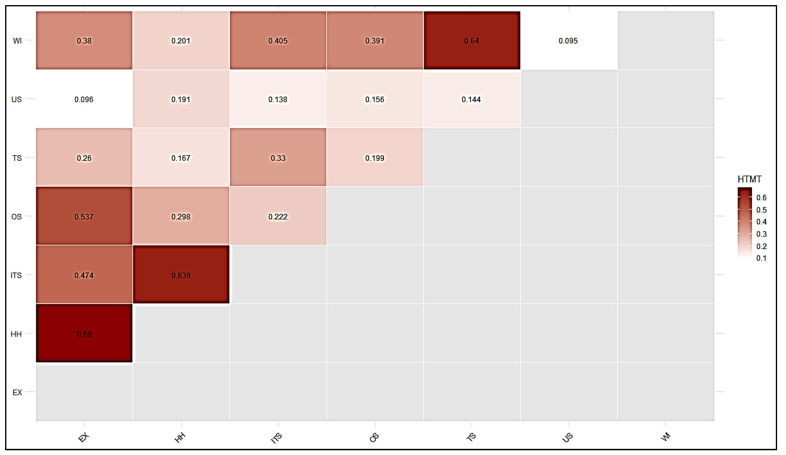
HTMT criterion matrix for second-order constructs. Off-diagonal values represent heterotrait-monotrait ratios. Discriminant validity is supported when HTMT values remain below the recommended thresholds (0.85 or 0.90).

**Figure 6 ijerph-22-01856-f006:**
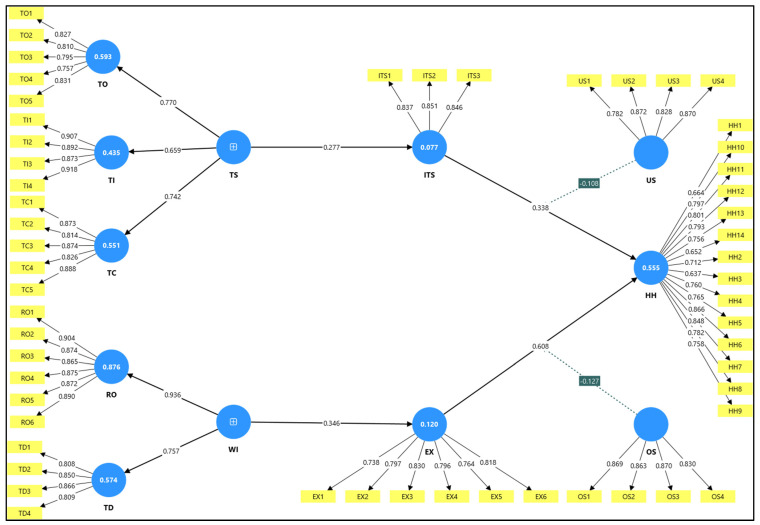
Measurement Model.

**Figure 7 ijerph-22-01856-f007:**
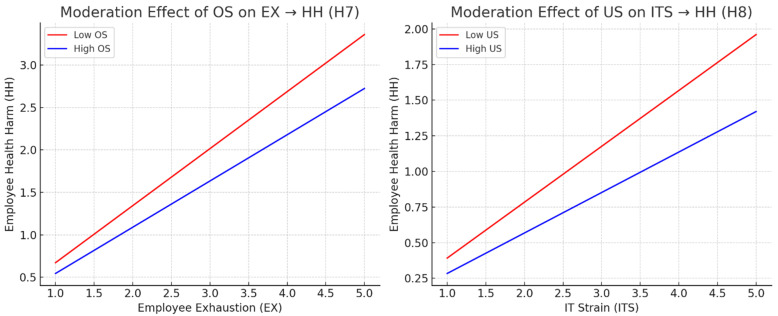
Moderation Effect.

**Figure 8 ijerph-22-01856-f008:**
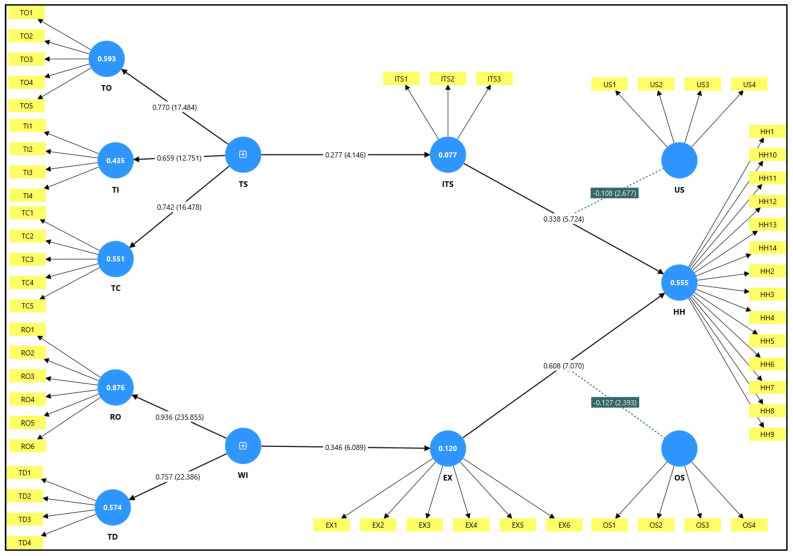
Structural Model (Path coefficient and t-Values).

**Table 1 ijerph-22-01856-t001:** Summary of Participants.

Demographic Values	Categories	Frequency	Percentage
Gender	Male	189	75.6
	Female	63	24.4
Age	Under 30 Years	6	2.38
	31–45 Years	30	12
	46–60 Years	191	75.79
	60+ Years	25	10
Education	Bachelor’s	48	19.05
	Master’s	191	75.79
	Doctorate	13	5.16
Employment	Managerial	160	63.5
	Mid-Level Employees	92	36.5
Service	6–10 Years	7	2.8
	11–15 Years	27	10.71
	16–20 Years	185	73.41
	20+ Years	33	13.09

**Table 3 ijerph-22-01856-t003:** HTMT-DV (1st-Order Construct).

	EX	HH	ITS	OS	RO	TC	TD	TI	TO	US
EX										
HH	0.68									
ITS	0.474	0.639								
OS	0.537	0.298	0.222							
RO	0.38	0.207	0.45	0.316						
TC	0.045	0.067	0.097	0.08	0.246					
TD	0.249	0.121	0.189	0.38	0.532	0.181				
TI	0.393	0.2	0.391	0.273	0.835	0.226	0.483			
TO	0.149	0.103	0.246	0.09	0.374	0.398	0.208	0.335		
US	0.096	0.191	0.138	0.156	0.08	0.099	0.087	0.109	0.102	

**Table 4 ijerph-22-01856-t004:** HTMT-DV (2nd-Order Construct).

	EX	HH	ITS	OS	TS	US	WI
EX							
HH	0.68						
ITS	0.474	0.639					
OS	0.537	0.298	0.222				
TS	0.26	0.167	0.33	0.199			
US	0.096	0.191	0.138	0.156	0.144		
WI	0.38	0.201	0.405	0.391	0.64	0.095	

**Table 5 ijerph-22-01856-t005:** Path Analysis (SMA Results).

Hypothesis	Path	Beta Value	SD	t Value	*p* Value	LLCI	ULCI	Decision
H1	WI ≥ EX	0.346	0.057	6.089	0	0.249	0.437	Supported
H2	EX ≥ HH	0.608	0.086	7.07	0	0.482	0.77	Supported
H3	TS ≥ ITS	0.277	0.067	4.146	0	0.17	0.389	Supported
H4	ITS ≥ HH	0.338	0.059	5.724	0	0.233	0.435	Supported
H5	WI ≥ EX ≥ HH	0.21	0.044	4.735	0	0.145	0.286	Supported
H6	TS ≥ ITS ≥ HH	0.094	0.026	3.662	0	0.057	0.141	Supported
H7	OS × EX ≥ HH	−0.127	0.053	2.393	0.009	−0.226	−0.047	Supported
H8	US × ITS ≥ HH	−0.108	0.04	2.677	0.004	−0.172	−0.036	Supported

**Note:** Technostress: TS; IT Strain: ITS; Employee Health Harm: HH; Work Intensification: WI; Employee Exhaustion: EX; Organizational Support: OS; and User Satisfaction: US.

**Table 6 ijerph-22-01856-t006:** R-square of Endogenous Constructs.

Construct	R^2^	Effect
EX	0.12	Weak
HH	0.555	Moderate
ITS	0.077	Weak

Note: Model explains a moderate to large amount of variance in health harm. It suggests that a dual stressor is effective in predicting health harm outcomes.

**Table 7 ijerph-22-01856-t007:** Effect Size of Path Coefficients.

Predictor → Outcome	f^2^	Effect
WI ≥ EX	0.136	Small to Medium
EX ≥ HH	0.306	Medium to Large
TS ≥ ITS	0.083	Small
ITS ≥ HH	0.203	Medium

Results highlight exhaustion is the strongest driver within the model.

**Table 8 ijerph-22-01856-t008:** Summary of Hypothesis Testing.

	Hypotheses	Decision
H1	Work intensification has a significant positive effect on employee exhaustion.	Supported
H2	Employee exhaustion has a significant positive effect on employee health harm.	Supported
H3	Technostress has a significant positive effect on IT strain.	Supported
H4	IT strain has a significant positive effect on employee health harm.	Supported
H5	Employee exhaustion mediates the relationship between work intensification and employee health harm.	Supported
H6	IT strain mediates the relationship between technostress and employee health harm.	Supported
H7	Organizational support moderates the relationship between employee exhaustion and health harm such that the relationship is weaker when organizational support is high.	Supported
H8	User satisfaction moderates the relationship between IT strain and employee health harm such that the relationship is weaker when user satisfaction is high.	Supported

**Note:** Technostress: TS; IT Strain: ITS; Employee Health Harm: HH; Work Intensification: WI; Employee Exhaustion: EX; Organizational Support: OS; and User Satisfaction: US.

## Data Availability

The original contributions presented in this study are included in the article. Further inquiries can be directed to the corresponding author.

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
