# Peer review of "Twin Threats in Digital Workplace: Technostress and Work Intensification in a Dual-Path Moderated Mediation Model of Employee Health"

_ijerph, 2025, doi:10.3390/ijerph22121856_

Round 1
Reviewer 1 Report
Comments and Suggestions for Authors
Dear author(s),
Thank you for the opportunity to read your manuscript and get acquainted with the research results you report. I find the text interesting, with potential contribution to relevant literature. However, it needs some more work to make it publishable.
I have a number of comments, which I hope could give some guidance on how to improve the text.
Major comments:
1) Section 3.2 reports rather high Cronbach’s alpha especially for the scales consisting of large number of items (Employee Health Harm: 14, Work Intensification: 10, Employee Exhaustion: 6). High Cronbach alpha (above 0.95) may indicate issues with the scale, such as some of the items are redundant, etc. that need to be commented in the text. Using Composite Reliability measure in addition to alpha could help reconfirm the reliabilities. When performing SEM, CR is often seen as better suited reliability criterion. You may consider other options, such as splitting the scales into their subscales, which you already mention in the text (such as for Work Intensification and Employee Exhaustion).
2) In Section 3.4 it is advisable to argument your choice of missing values imputation method. There are various methods, and you need to first assess the type of missing values in order to decide which method is best.
3) (a) Section 4.1.1, Table 2 is difficult to evaluate – the constructs abbreviations should be explained. Now it is difficult to track & compare with the reliability checks reported in Section 3.2.
(b) Lines 757-758 introduce for the first time in the manuscript “first-order and second-order constructs” which were not mentioned earlier. If you work with second-order constructs, you should explain why and how do you form the second-order constructs before presenting the results. Consider also my 1st comment above.
4) The Discussion section could be enriched by pointing at how your results support or differ from previous findings, and especially the part that talks about theoretical contributions.
Minor comments regarding:
1) Editing. Careful proofreading and editing of the whole manuscript may help improve the text, such as use properly punctuation in “Modern work places now combine automation, robotics cloud platforms and AI … “ (lines 47-48) and “The mediating constructs in this model emotional exhaustion and IT strain are conceptualized as …”(lines 146-147); improve grammar in “..that chronic demands steadily erode resource pools if not replenished.” (lines 144-145); add missing words in “…leading to health unless buffered by user satisfaction and organizational support.” (lines 163-164), Figures 6, 7, and 8 are not introduced in the text, etc.
2) References. When stating “Similarly, few others demonstrate that …” (line 268), the reader would expect to see references (who these “few others” and what & in which studies they “demonstrate that”). Other examples are “Empirical studies support this meditating pathway. Researchers found that…” (lines 462-463), etc.
3) Abbreviations. A phrase or a term is abbreviated only the first time it appears in the text. Afterwards, use only the abbreviation or the full term. Now, the abbreviations of Conservation of Resources (COR) Theory § Technological Work Burnout (TWB), for example, are used in combination with the full term at almost each instance they appear in the text.
4) Conciseness.
(a) The text above Table 1 (Section 3.6) basically reiterates what is already visible in Table 1. You may consider editing the text to extract what is important form the tables / comment the tables, otherwise it is redundant. The same comment is valid for the text above Table 5 (Section4.2.1)
(b) Figure 2 also reiterates what is already visible in Table 1. Please choose which form of presenting the data you would like to keep, and rather delete the other one. The same comment is valid for Figure 3 & Table 2, Table 3 & Figure 4, and Table 4 & Figure 5.
Author Response
Respected Reviewer
Thank you very much for sharing your observations. I made changes as desired, please. My point to point response is attached in a PDF file
For perusal, please
Jawad

Reviewer 2 Report
Comments and Suggestions for Authors
Several areas require substantial improvement in structure, clarity, and focus.
Major Comments
-
Theoretical Contribution
The paper would benefit from a clearer articulation of its novelty. Please explain explicitly how the dual-path model extends prior applications of the Conservation of Resources (COR) theory and why this integration of technostress and work intensification provides new theoretical insight. -
Structure and Readability
The discussion is overly long and repetitive. Consider reorganizing it into concise subsections: Theoretical Implications, Practical Implications, Limitations, and Future Research. This will improve clarity and reader engagement. -
Connection to Results
Strengthen the link between empirical results and theoretical interpretation. Cite key statistical values (e.g., β, p, R²) when explaining each finding to demonstrate analytical coherence. -
Practical and Policy Implications
Recommendations for managers and policymakers are too general. Provide specific, actionable suggestions—for example, digital ergonomics programs, IT user training, or psychosocial risk management practices—to increase applied relevance. -
Currency of Literature
Update the reference list with recent studies (2023–2025) on digital well-being, technostress, and Industry 4.0. This will demonstrate engagement with the latest developments and strengthen the paper’s contemporary value. -
Limitations and Future Directions
Add a dedicated subsection acknowledging methodological and contextual limitations (cross-sectional design, self-report data, cultural factors). Propose clear avenues for future research. -
Conclusion
End with a concise and impactful statement summarizing the main contribution—emphasizing the dual nature of technological and organizational stressors and the importance of resource-protective strategies.
Author Response
Respected Reviewer
Thank you for sharing observations. I made changes accordingly. Point to point reply is attached in a PDF file
Regards
Jawad

Reviewer 3 Report
Comments and Suggestions for Authors
1- The methodology includes good details on sampling, instrument, and SEM analysis. However, it could further clarify questionnaire validation, sampling rationale, and ethical considerations beyond the brief note at the end. Including reliability/validity testing tables in the main text would strengthen replicability.
2- Figures are informative (e.g., SEM diagrams, demographic charts), but some could benefit from higher resolution and consistent formatting (especially Figures 5–8). A clearer legend and uniform labeling style (font, alignment) would improve presentation quality for IJERPH standards
Author Response

(The authors gave the same response as above.)

Reviewer 4 Report
Comments and Suggestions for Authors
This manuscript examines how technostress and work intensification jointly affect employee health through mediating and moderating mechanisms. The topic is highly relevant to digital transformation and occupational health. The paper is clearly organized and employs validated measures and PLS-SEM analysis. However, several theoretical and methodological aspects could be refined to improve coherence, validity, and the strength of interpretation.
Major comments:
- The Conservation of Resources (COR) theory is repeatedly cited across sections, often with similar phrasing. Acronyms such as Technological Work Burnout (TWB) and related constructs are also used inconsistently. The theoretical background should be condensed and terminology streamlined, with clearer distinctions among overlapping constructs (IT strain, exhaustion, burnout). The authors are encouraged to highlight how this study extends COR theory in digital manufacturing contexts rather than reiterating its general assumptions.
- The hypotheses are expressed in causal terms (“leads to,” “results in”) although the research design is cross-sectional. The authors should rephrase these hypotheses using associative wording (“is related to,” “is associated with”) and explicitly note that causal inference cannot be drawn from the present data.
- With 252 participants and several mediators and moderators, the sample lies at the lower bound for complex PLS-SEM models. The authors are encouraged to report a power analysis or justify sample adequacy using accepted criteria to demonstrate that statistical power is sufficient.
- All hypothesized paths are significant, which is uncommon in models of this complexity. The discussion should comment on the relative strength and practical relevance of these effects (R², f²) rather than focusing solely on significance levels. Including a more critical interpretation would enhance credibility.
- Figure 1 would be clearer if the hypothesis numbers (H1–H8) were indicated along the corresponding paths, helping readers connect the conceptual model with the hypotheses testing section.
- The manuscript briefly references the Job Demands–Resources (JD-R) framework alongside Conservation of Resources (COR) theory. However, this mention is not clearly integrated into the overall theoretical reasoning. Clarifying whether the reference to JD-R serves as a complementary perspective or a supporting citation would improve theoretical consistency.
- As all measures were self-reported at a single time point, the possibility of common-method variance should be acknowledged or statistically tested (e.g., Harman’s single-factor test).
- The demographic distribution (Figure 2) indicates that most participants were male, mid-to-senior-level employees aged 46–60 years. While this composition reflects the reality of the studied industrial sector, it may also limit the extent to which the findings can be generalized to younger or non-managerial workers. Acknowledging this contextual specificity would enhance the transparency and interpretive scope of the results.
- The practical implications section is well written and offers relevant recommendations. Nevertheless, its assertive tone exceeds what can be supported by a cross-sectional correlational design. These implications should be framed as potential applications rather than demonstrated outcomes, ensuring that managerial claims remain consistent with the empirical evidence.
- The limitations section correctly lists major constraints (self-report, cross-sectional design, cultural context) but remains descriptive. Discussing how these factors might influence the observed relationships would enhance transparency. A brief justification of sample adequacy through a power analysis is still recommended.
- Since the data originate from Pakistan’s manufacturing sector, a short reflection on how local work norms or digitalization policies might shape technostress perceptions would strengthen external validity.
Minor comments
- The tables and figures should include footnotes explaining all abbreviations and constructs (e.g., EX = Exhaustion, HH = Health Harm, ITS = IT Strain, etc.). As currently presented, several tables (e.g., Table 3 – HTMT) are not self-contained and require cross-referencing with the text to understand the variables. Providing full legends would align the manuscript with MDPI’s formatting standards for clarity and reproducibility.
- Several labels and numerical values in Figure 2 and 3 are too small or partially obscured. Increasing font size and simplifying layout would improve visual clarity and compliance with journal standards.
- Some visualizations duplicate information already shown in tables (for example, the correlation figures). A reduced number of more interpretive figures could streamline presentation.
- Certain paragraphs repeat theoretical explanations. Shortening and simplifying these sections would make the paper more readable without sacrificing rigor.
- While the theoretical background is solid and grounded in established frameworks, several of the references are relatively dated, with limited inclusion of studies published after 2020. Considering the rapid evolution of research on technostress, digital transformation, and employee well-being (especially in post-pandemic contexts), incorporating more recent empirical and review papers would strengthen the manuscript’s theoretical currency and relevance.
Author Response

(The authors gave the same response as above.)

Round 2
Reviewer 1 Report
Comments and Suggestions for Authors
Dear authors,
Thank you for your efforts to revise the text, its revised version is a significant improvement.
I have a couple minor comments:
1) Subsection 3.4 – if you did not impute missing variables, you can completely skip that subsection, and especially the first 2 sentences. Sufficient to say you used only completely filled-in questionnaires
2) Duplicating information in tables and figures – I still think this is redundant and does not help the reader to follow your line of thought. But, of course, this is your decision.
Wishing you success with your research.
Author Response
Respected Reviewer
Thankyou very much for further guiding me. I have incorporated your observations
Best Regards
Jawad

Reviewer 2 Report
Comments and Suggestions for Authors
Some theoretical explanations—particularly those related to the Conservation of Resources (COR) theory—are repeated across multiple sections. Streamlining these passages would enhance clarity and avoid redundancy, ensuring a more concise and focused theoretical narrative.
Comments on the Quality of English LanguageThe manuscript would benefit from a final round of professional English proofreading. Although the content is clear, several sentences are overly long or stylistically heavy, and minor grammatical inconsistencies remain throughout the text. These refinements would improve readability and alignment with journal standards.
Author Response
Respected Reviewer
Thankyou very much for further guiding me. I have incorporated your observations. Point by point document is attached for your perusal, please.
Best Regards
Jawad

Reviewer 4 Report
Comments and Suggestions for Authors
The authors have addressed the reviewer’s comments carefully, and the manuscript has significantly improved in both structure and coherence. At this stage, my remaining suggestions are purely related to visual clarity and minor wording, and do not concern the core theoretical or methodological contribution of the study.
Figure 1: Although the proposed model is described in the text as a dual-path moderated mediation, this is not immediately apparent from the visual representation. At first glance, the figure appears to consist mainly of direct paths between constructs, and the mediating and moderating mechanisms are not clearly distinguishable.
To further improve clarity, it would be useful to visually differentiate the types of relationships (e.g., mediation vs. moderation vs. direct effects), for example through different line styles (solid vs. dashed), symbols, or a short explanatory note in the legend specifying which hypotheses correspond to the mediating paths and which represent the moderating effects. In addition, the label should be changed to “Figure 1. Proposed Model” rather than “Theoretical Framework.”
Methods section: It may be worth briefly clarifying which analyses were conducted using PLS-SEM and which were conducted in other software (e.g., SPSS), as several programs are mentioned across different sections. A short clarification of the analytical workflow would help avoid potential confusion for the reader.
Author Response

(The authors gave the same response as above.)

Round 3
Reviewer 2 Report
Comments and Suggestions for Authors
1. Ethical approval timing
The manuscript states that data collection took place in June–July 2025, whereas the ethics approval was issued in August 2025. This is a serious concern, as research involving human participants must receive ethics approval before data collection begins. The authors must clarify this discrepancy or provide documentation explaining the institutional procedure. Without a satisfactory clarification, publication may not be possible.
2. Language clarity and style
While the manuscript is generally understandable, it contains numerous instances of awkward phrasing, overly long sentences, redundancies, and occasional grammatical errors. A thorough editing by a proficient English speaker is needed to improve readability and ensure professional academic tone.
3. Redundancy in the theoretical section
The background section, particularly the discussion of COR theory and related constructs, is excessively long and sometimes repetitive. Streamlining this section would improve flow and sharpen the theoretical contribution.
4. Citation inconsistencies and typographical errors
Several citation formats are inconsistent or contain errors (e.g., “[5656]”, “techno-overlord0”). The reference list and in-text citations should be carefully checked and corrected for accuracy and formatting.
Minor Issues
-
The “Health Harm” construct would benefit from a clearer operational definition, including explicit mention of the instrument used.
-
The demographic profile of the sample (mostly male, aged 46–60) limits generalizability; the authors acknowledge this but may consider reinforcing the discussion.
-
Some figures and tables could be introduced more smoothly in the narrative.
The manuscript would benefit from a final round of professional English proofreading. Although the content is clear, several sentences are overly long or stylistically heavy, and minor grammatical inconsistencies remain throughout the text. These refinements would improve readability and alignment with journal standards.
Author Response
Respected Reviewer!
I made an effort to address all your observations
Please find attached point by point response
Revised manuscript also attached at the required place
Best Regards
Jawad

Round 4
Reviewer 2 Report
Comments and Suggestions for Authors
Several theoretical explanations—particularly the Conservation of Resources (COR) Theory—are repeated across multiple sections.
Definitions of constructs are unnecessarily long and include repeated concepts.
The introduction could be more concise and focused.
Multiple in-text citations contain formatting problems (e.g., “[11 ]”, “[13,]”).
The reference list does not fully match the in-text citation order, and some items seem missing or incomplete.
All references must be fully aligned with MDPI’s citation style.
The manuscript contains template placeholders such as “Academic Editor: First name Last name” and incomplete metadata fields.
Some tables and figures are not aligned, appear low-resolution, or are inconsistently formatted.
Captions and numbering do not always follow MDPI guidelines.
Some demographic results appear unusual or require clarification:
-
75.79% of respondents are 46–60 years old.
-
All respondents are reported as married.
-
73.41% have 16–20 years of experience.
These distributions are atypical for a manufacturing sector sample and should be explained, justified, or checked for accuracy.
Some paragraphs combine several unrelated ideas, reducing clarity.
The theoretical framework section is dense, with overlapping arguments.
Definitions of variables should be streamlined and made more precise.
There are inconsistencies in terminology (e.g., “techno-overlord” vs. “techno-overload”).
Ensure consistent use of construct names throughout the manuscript.
Minor Comments
There are multiple issues with spacing, line breaks, and alignment of text.
Ensure uniform formatting according to MDPI standards.
Some abbreviations (e.g., TWB, JD-R) are introduced without proper first-use definition.
Please verify that all abbreviations are defined at first mention.
Tables and figures should be placed immediately after first reference in the text.
Some figures appear blurry and may need higher resolution.
Although the SEM analyses are complete, the reporting of some statistics should follow a consistent format (e.g., uniform decimal places, consistent use of t-values and confidence intervals).
The use of purposive sampling should be better justified, and any limitations clearly acknowledged.
Comments on the Quality of English LanguageThe manuscript would benefit from a final round of professional English proofreading. Although the content is clear, several sentences are overly long or stylistically heavy, and minor grammatical inconsistencies remain throughout the text. These refinements would improve readability and alignment with journal standards.
There are frequent grammatical errors, misspellings (e.g., rapidily, framwork, overlord instead of overload), awkward phrasing, and overly long sentences.
gain, a professional language edit is strongly recommended to ensure clarity and academic style.
Author Response
Respected Reviewer
Point by point response is attached for perusal, please
Best Regards
